# High-Temperature Corrosion of APS- and HVOF-Coated Nickel-Based Super Alloy under Air Oxidation and Melted Salt Domains

**DOI:** 10.3390/ma14185119

**Published:** 2021-09-07

**Authors:** Ibrahim A. Alnaser, Mohammed Yunus, Rami Alfattani, Turki Alamro

**Affiliations:** 1Mechanical Engineering Department, King Saud University, Riyadh 11421, Saudi Arabia; ianaser@ksu.edu.sa; 2Department of Mechanical Engineering, Umm Al-Qura University, Makkah City 24372, Saudi Arabia; rafattni@uqu.edu.sa (R.A.); tsamro@uqu.edu.sa (T.A.)

**Keywords:** alloy 80A, high-velocity oxy-fuel, atmospheric plasma spraying, hot corrosion resistance, molten salt environment, air oxidation

## Abstract

Various thermal spraying approaches, such as air/atmospheric plasma spraying (APS) and high-velocity oxy-fuel (HVOF) spraying, are widely employed by plants owing to their flexibility, low costs and the high surface quality of the manufactured product. This study focuses on the corrosion behavior of a Ni superalloy coated with powder Cr_3_C_2_-25NiCr through APS and HVOF at 950 °C under air oxidation and Na_2_SO_4_ + 0.6V_2_O_5_ molten salt environments (MSE). The results show that HVOF-deposited Ni superalloys have higher hardness and bond strength than the respective APS coating. The thermo-gravimetric probe reveals that the Ni superalloys exposed to an oxidizing air environment has a minor mass gain compared to those under the MSE domain for both non-coated and coated samples, in line with the parabola curvature rate oxidizing law. The Ni superalloys show good corrosion resistance but poor oxidation resistance in APS-deposited Ni superalloys under the MSE. HVOF-coated Ni superalloys in both environments exhibit better corrosion resistance and lower mass gain than APS-coated superalloys. The excellent coating characteristics of HVOF-coated Ni superalloys lead to their better high-temperature corrosion performance than APS.

## 1. Introduction

Presently, for running power plants (PP), it is mandatory to follow strict emission conditions to ensure environmental safety. Harmful emissions such as acid rain, severe air pollution and global warming can be reduced in power plants by using advanced ultra-supercritical technology (AUST). AUST generates more power and less volume of effluent emissions (CO_2_, NO_x_ and SO_2_) for the same fuel consumption and capital investment when compared to subcritical (conventional) PPs. The conventional plant efficiency is 0.85% more than AUST. The AUST-PP needs higher operating values, such as 365 bar and 720 °C, than supercritical PPs, with inlet conditions of 240 bar and 565 °C [1]. The superalloy materials as structural elements of these PPs need to withstand hot corrosion (HC) at high operating conditions. HC involves high-temperature (HT) oxidation, which decomposes materials (metals and alloys) in the presence of salt or molten salt deposited on its surface. Molten salt will degrade materials faster than the air/gaseous oxidation conditions. The increased porosity and developed destructive oxide scale on the surface and sulfides in the substrate are an indication of HC attack occurring at HTs [2]. The HC problem has been identified in boilers, industry waste fireboxes and incineration engines. Kamal et al. [3] explained the HC phenomenon as dispersion of the defensive oxide deposit by reaction of fused salt (Na_2_O). As the gas turbine (GT) working temperatures are very high, it is necessary to obtain newer materials and plan improvements to take into account the higher efficiency and power production compared to conventional GTs. Meanwhile, the higher intake gas temperature in GT systems makes them more prone to HC, especially in the presence of contaminants such as Na, V and S in fuel-mode salt residues on the constituent surface [4]. The HC phenomenon in boilers remains analogous to GTs [5]. The use of magnesium and magnesium-based inhibitors can prevent HC to a certain degree [6] and widespread probes revealed that Mg, Mn and Ca-based inhibitors can effectively reduce HC in a Na_2_SO_4_ + 0.6V_2_O_5_ environment at 900 °C, but their use is hindered because of practical difficulties in feeding the inhibitors into the fuel [7].

Most advanced nickel (Ni) superalloys contain Cr and Al content in an adequate quantity to cause the particular oxidation of these elements, which are promising GT materials. Ni superalloys produce either alumina or chromia. Ni superalloys with high Cr are suitable for environments with sulfur and carbon. Alloy 80A (UNS NO7080 (ASTM B637-06)) is a Ni superalloy fabricated for HT applications at 815 °C, primarily composed of Ni and Cr and strengthened by Ti, Al and C addition. Alloy 80A is mostly employed in the components (blades, combustion liners, transition pieces, seal rings, joining bolts and discs) of GTs and steam turbines (ST) [1]. The efficiency of STs and the lifetime of their parts can be improved by the selection of an appropriate material and production technique.

However, further research work is needed to ensure the reliable corrosion behavior of the different materials and coating powders in various environments. It is necessary to take into account the inherent characteristics of thermal spray coatings (TSC) [8]. High-temperature corrosion is inevitable in the presence of an aggressive environment, particularly in fabricated parts exposed for long durations to high-temperature environments [9]. There are two different methods used to protect the long-lived components and enhance the corrosion resistance: the addition of Al, Cr and Ti with a small % of Y, Zr and H_f_ to the base alloy [10] and surface modification technologies for depositing protective coatings. These techniques include diffusion coating, TSC, chemical and physical vapor deposition (CVD and PVD), weld overlay, etc. [11]. Many considerable efforts have been devoted to investigating the properties of components with and without coatings when exposed to different environments. However, our understanding the corrosive/protective mechanisms of different substrates/coatings is limited and they need further study. When thick coatings were applied to boiler and turbine components, their lifetimes were enhanced under severe working conditions [12]. The initiation of hot corrosion requires a longer time than its propagation.

The significant benefits of TSC are high production rates for coating thicknesses above 100 µm, applicable for a wide range of substrate (metal and non-metal) and coating powders (ceramic and carbide) [13,14]. However, additional research work is needed to ensure the reliable corrosion behavior of the multiple materials with different coating powders in various environments. The oxidation resistance (OR) of nickel superalloy at 950 and 1000 °C for 140 h in an air environment and the oxidation behavior of the alloy followed the parabola curvature rate oxidizing law at 950 °C and slightly deviated from it at 1000 °C [15]. There is no scale spallation from the alloy at 950 °C but minor spallation of scales was noted at 1000 °C [16]. There are two types of HC processes: type I is high-temperature HC from 850 to 950 °C caused by Na_2_SO_4_ fused salt because of its high thermodynamic stability, and type II (low-temperature (LT)) HC at 650 to 800 °C occurs in the eutectic mixtures of fused salts. The basic fluxing is the primary HC mechanism occurring in alumina and chromia, forming nickel-based superalloys. The combustion of low-quality graded fuels in power generation systems leads to the occurrence of vanadium contaminants, which oxidize and form V_2_O_4_ and V_2_O_5_ [17]. High corrosion above 800 °C is found initially in all materials, but later, this corrosion becomes constant or lowers with a further increase in temperature. The formed Na_2_CrO_4_ is mixable in Na_2_SO_4_ melt. Unalloyed Ni material exhibits less HC resistance than Ni-Cr, Ni-V and Ni-Mo alloys in the presence of fused Na_2_SO_4_ salt.

Coatings applied with various deposition methods, such as wire arc, atmospheric or air plasma (APS), high-velocity oxy-fuel (HVOF), cold spraying, etc., possess different characteristics [7]. APS coating during the HT oxidation of Ni superalloys (Superni 75, 600, and 601) coated with Ni-20Cr powder at 900 °C revealed that the coating was retained without spallation for up to 50 cycles [18]. The influence of a Cr_3_C_2-_NiCr coating using APS during the HT oxidation of 2.25Cr-1Mo steel in a constant-temperature trial at 700 °C for three days and weight loss were assessed at several time points. The results showed an enhancement in the OR of 2.25Cr-1Mo steel as the actual OR improved in the Cr_2_O_3_ layer formed by the C_r3_C_2_NiCr (chromium carbide–nickel chromium) coating [19,20]. The corrosion diffusion mechanism in an APS-sprayed Cr_3_C_2_-NiCr coating through the inward diffusion of oxygen anions from the oxidation environment to coating layers was evaluated. In the non-coated base metal, the outward migration of iron cations from the substrate surface occurred. Two different diffusion mechanisms of corrosion improved the OR to a significant range. The HVOF process uses gases such as hydrogen, acetylene and propylene and liquid fuels such as kerosene to melt the high powder particle velocities from 480 to 840 m/s and deposit them onto base metals. During the HC evaluation of a T-91 alloy steel boiler tube and HVOF deposited with Cr_3_C_2_ + 0.25 Ni20Cr under 540, 700 and 850 °C temperatures, the coating showed the most negligible weight gain compared to uncoated specimens as deposition reduced the corrosion frequency significantly [21]. HC deportment of base metals (SA213-T22, MDN310 and Superfer800H) and HVOF deposited with Cr_3_C_2_-0.35NiCr + 0.05Si powder under (Na_2_SO_4_ + 0.60V_2_O_5_) at 700 °C showed a laminar structure with low porosity [22,23]. Steel base metals exhibited a better OR than uncoated steel. Both coated and uncoated steel followed the parabola curvature rate oxidizing law, and the parabolic rate constant of coated steel was lower than that of uncoated steel [24]. Cr_2_O_3_ and SiO_2_ oxides developed over the exterior layer were inactive in the melted NaVO_3_ salt.

A limited number of studies are available on the HT corrosion behavior of Ni superalloy exposed to air oxidation/corrosion (AC) and a melted salt environment (MSE) at 950 °C. As the temperature is quite high, an effective method required to extend this study is to apply a protective coating that resists the corrosion of the exposed surfaces. Such deposits have been found to improve the longevity of the base material/alloy to a certain level [25,26,27,28]. Numerous TSC methods, such as APS and HVOF, are extensively utilized to deposit high Ni and Cr powder coatings onto various PP component surfaces in order to withstand HC [4,29]. The present work utilizes Cr_3_C_2_-25NiCr powder, which was chosen as the high chromium (Cr) content of the material is advantageous for providing HC resistance by protecting the oxidic layers of Cr_2_O_3_ on the Ni superalloy’s surface. Thermally sprayed Cr_3_C_2_-25NiCr depositions are extensively employed in several areas to protect the exposed surfaces and resist wear and HT corrosion under harsh conditions [30]. The coatings contribute exceptional tribological attributes at raised temperatures, improve the components lifetimes at such HT conditions in the PP system, and sustain high wearing resistance up to 880 °C in turbine and boiler components [31]. Ni-Cr-based metal coatings can develop remarkable HC resistance in AC. However, there are two different oxidation conditions (such as AC and MSE) for two different TSC methods. Hence, Na_2_SO_4_ + 0.6V_2_O_5_ salt is chosen for the MSE, employing thermocycling terms associated with real-time automated background climate specifications, where the production plant disorder occurs frequently.

From the above explorations, it can be inferred that earlier researchers have not concentrated on the mechanical characteristics and oxidation resistance of Cr_3_C_2_-25NiCr depositions at high temperatures by means of modern TSCs under air and melted salt. The present work examines the HVOF and APS coating characteristics, such as hardness and bond strength, for Cr_3_C_2_-25NiCr powder and the HC behavior of uncoated alloy 80A in AC and MSE. Moreover, we investigate the HC characteristics of the deposited Ni superalloy’s surface by means of APS and HVOF spray processes when exposed to 950 °C for 50 cycles.

## 2. Experimental Procedure

The experimental procedure adopted to deposit the coatings via the APS and HVOF methods using Cr3C2-25NiCr powder to determine the HC resistance at 950 °C under AO and MSE is illustrated in detail. The procedure to characterize the coatings includes measurement of the hardness, bond strength and HC resistance during both processes, as explained in detail. The base metal material, Ni superalloy (commercial name: alloy 80A), is utilized for this current research. The chemical composition of the Ni superalloy as the as-received base metal is detailed in Table 1. The Ni superalloy selected had the shape of a 64-mm-diameter cylindrical rod, sliced into 64 mm × 7 mm thick discs and sliced into rectangular pieces of 20 mm × 15 mm × 7 mm by employing the Electrical Discharging Machining (EDM) process. The coating was applied on the base metal alloy 80A using APS and HVOF methods in the present study. Before coating, the base metals were initially grit-blasted with white alumina (Al_2_O_3_) at 4 bar blasting pressure and then glazed utilizing fine-sized grit silicon carbide (SiC) paper [32].

The two coating methods (refer to Figure 1 and Figure 2) were used to deposit the coatings onto to all the surfaces of the Ni superalloy utilizing APS (thickness of 150 µm) and HVOF (200 ± 5 µm) processes; the employed process parameters are listed in Table 2. For the APS method, we deposited the coating via a 3MB plasma air gun (Sulzer Metco made), with 100 kW power input, and we used argon (Ar) and hydrogen (H_2_) as input gases. The HVOF coating was performed using a multiple coating system operated with oxygen and jet fuel [33].

### 2.1. Characterization of Coatings

Various important characteristics of the coated substrates were evaluated, including micro-hardness, bond strength and HC resistance.

#### 2.1.1. Bond Strength Testing

The tensile-based adhesive/bond strength of the coating was evaluated independently utilizing a novel kind of specimen holder on a developed cylindrical-shaped dummy-coated specimen with a diameter of 25.4 mm × length 38.1 mm.

All dummy surfaces were roughened. Every roughened dummy face was fixed on top of a substrate sample using ‘epoxy 900-C’ (polymeric adhesive) and subjected to tension after being mounted on the jig. Specimens were placed in a universal testing machine and pulled with a crosshead speed of 0.02 mm/s until the two parts separated. The bond strength of the coatings was estimated considering ASTM C633 standards. When pulled in a plane, or once rupture occurred, the maximum load was recorded. Bond strength is the ratio of the maximum load to the cross-section area of the samples. After thermal curing with epoxy glue, the deposited and undeposited samples were placed between pairing cylinders of 50 KN capacity. The schematic diagram for the bond strength testing is shown in Figure 3. The average value was estimated, excluding the maximum and minimum value, in each set of five readings to eliminate any possible distortion of the results owing to any outliers. This approach was taken with all tests.

#### 2.1.2. Pictorial Examination

After each cycle, every sample was examined visually for any change in color, crack formation and changes in surface properties. After the completion of 50 cycles, each specimen was examined, and their macrographs were prepared. The physical visual examination (spallation and formation of cracks) on the surface of sample was performed using the macrographs.

#### 2.1.3. Mass Gain Analysis

The kinetics of hot corrosion were determined by the measurement of the mass change values of each sample after each cycle. Then, the graph was plotted between the mass change data and number of cycles.

#### 2.1.4. Cyclic High-Temperature Corrosion Test

A trial of HT corrosion was performed for up to fifty cycles on the undeposited and APS- and HVOF-deposited Ni superalloys under both AO and MSE (Na_2_SO_4_ + 0.6V_2_O_5_) environments. Specimens were heated in a tubular furnace at 905 °C for around 60 min and then cooled for 20 min to reach an ambient temperature—this was considered one cycle. First, a non-deposited base metal, as well as the APS- and HVOF-deposited samples, were heated at 200 °C in the furnace (refer to Figure 4) before being exposed to MSE to apply the Na_2_SO_4_ + 0.6V_2_O_5_ salt. Purified water with a salt mixture of Na_2_SO_4_ + 0.6V_2_O_5_, called molten salt, was applied at a uniform rate of 3.0–5.0 mg/cm^2^ on the base metal’s surface using the brush. Then, they were kept at 150 °C in the box furnace for 4 h to ensure proper adhesion of salt by removing the moisture [35]. The mass change in the Ni superalloy with alumina boat (AB) was measured for every cycle during the HC test together with fragments of deposited oxide scale in the AB. Tests were repeated five times to maintain accuracy and repeatability of measurements.

#### 2.1.5. Micro-Hardness Testing

Micro-hardness testing was conducted on cross-sections of the deposited base metals and it was measured from the base metal’s surface and the top surface of the deposition by taking nine indentations randomly, adopting ASTM E384 procedures [36].

## 3. Results and Discussion of Air Plasma Spray Coating

Ni superalloys were deposited with the APS and HVOF systems employing Cr_3_C_2_-25NiCr powder. Various coating parameters, such as micro-hardness and bond strength, were measured, and thermogravimetric analysis was performed, for every cycle for up to fifty cycles. Results were plotted to estimate the mass gain of the undeposited and deposited samples under AO and MSE oxidation environments.

### 3.1. Microhardness (H_v_) of the Base Metal and Depositions (APS and HVOF)

Figure 5 illustrates the micro-hardness profile across the alloy 80A and Cr_3_C_2_-25NiCr powder-deposited base metals using the APS and HVOF processes. The H_v_ was measured using a Vickers indenter on the polished coating section under a load of 300 g with a dwell time of 10 s for every hardness test. The average of nine readings on each sample was taken as a data point. The average micro-hardness observed on the undeposited base metal alloy 80A was 298 Hv_0.3_, and that of the deposited base metals was 840 Hv_0.3_ (APS-coated) and 862 H_V0.3_ (HVOF-coated). The hardness values increased by 172% and 178% for the APS- and HVOF-deposited base metals compared to the base alloy 80A. The enhanced hardness in the Cr_3_C_2_-25NiCr powder was due to the presence of the Cr_3_C_2_ hard phase in the NiCr matrix and the high kinetic energy generated by the powder particles [37].

### 3.2. Adhesion/Bond Strength of the Deposits

The bond/adhesion strength was noted as 41 and 72 MPa in the APS- and HVOF-deposited NI superalloys; it was between 34 MPa and 68 MPa according to the ASTM C633 standard [38], and the type of deposit failure was cohesion [39]. On the other hand, in HVOF depositions, glue-type failure occurred and the result was greater than the glue strength employed in the test.

### 3.3. Pictorial Examination

Figure 6a–f depict the macrographs of undeposited (refer to Figure 6a,b), APS- (refer to Figure 6c,d) and HVOF-deposited base alloy 80A (refer to Figure 6e,f) up to the 50th cycle under the AO and MSE environments (Na_2_SO_4_ + 0.6V_2_O_5_) at 950 °C. In the first few cycles (first to fifth), the color of the undeposited and APS-deposited Ni Superalloy in both environments changed to light and dark grey colors. These colors started to change to a brown color after the 20th cycle, and the same occurred up to the fiftieth cycle. Random scale accumulation was witnessed during the third cycle onwards in the MSE domain on both the undeposited and APS-deposited Ni superalloy. The severe ejection of fragments of coated powder/spallation at edges was observed in the APS-deposited Ni superalloy under the AO domain, but very minimal deposited layer spallation at the top was observed from the fifth cycle to the fifteenth cycle; after this, no spallation occurred in subsequent cycles. We observed the formation of bright spots on all the Ni superalloys after completion of the seventh cycle, and these continued to gradually increase in size and quantity in subsequent cycles up to the fiftieth cycle. Moreover, the corroded stock started paring off from the eighth cycle onwards, only under the MSE domain, in both the undeposited and deposited Ni superalloys [40].

The dark grey color of the HVOF-deposited Ni superalloys under the AO domain (Figure 6e) changed into a grey color during the eighth cycle, and then no color change was found till the last cycle. During the seventh cycle, the generation of white dots was perceived on the deposited Ni superalloys exposed to the AO domain. The deep brown shade of the deposited Ni superalloys (refer to Figure 6e) subjected to MSE changed to a grey shade during the twelfth cycle and remained the same until the fiftieth cycle. The presence of white spots started to emerge under the MSE substrate during the twelfth cycle. Crack formation and paring off of deposited layers were not seen in HVOF-deposited Ni superalloys till the last cycle. Spallation of scaling started from the beginning of the seventh cycle onwards in both the deposited and undeposited samples when exposed to the MSE domain (refer to Figure 6b,f) and proceeded till the fiftieth cycle. This showed that the scale formation occurred mainly in the MSE domain compared to Ni superalloys subjected to the AO domain (refer to Figure 6a,e).

### 3.4. Oxidation Kinetics

Referring to Figure 7, the plots of the Cr_3_C_2_25NiCr-deposited base metals show greater weight gain than the undeposited alloy 80A base metal when subjected to AO. In particular, the weight gain of the APS-deposited base metals in the AO domain was greater than all other cases. At every 50th cycle, the calculated total weight gain values (refer to Table 3) showed that AO in both (deposited and undeposited) base metals provided significantly less gain than under MSE domains. It was also noted that Cr_3_C_2_25NiCr powder deposited by APS on base metals led to greater weight gain in the MSE domain than in the undeposited Ni Superalloy. The MG of the APS-coated substrate in the MSE domain was higher than that of the HVOF-deposited base metals for the same corrosion domain.

The MG graph shows that mass progressively increased with the number of cycles in the AO domain. In contrast, in the case of MSE, the mass gain was greater at the beginning of some cycles and progressively increased in additional cycles. Figure 8 for (MG/Area)^2^ implies that the corrosion process conforms to the parabolic law for the corrosion rate for the AO domain, and there is a deviation from this law in the case of the MSE domain for the base metals of both undeposited and APS-deposited specimens. The parabolic rates (K_P_) are listed in Table 3. As seen in Figure 8 and Table 3, K_P_ values were high under the MSE domain for both undeposited and APS-deposited Ni superalloys. APS-deposited Ni superalloys under the AO domain had higher K_p_ values than undeposited ones when exposed to the same domain. The undeposited and APS-deposited substrates in both the AO and MSE environments conformed to the parabolic law for the corrosion rate (refer to Figure 8). K_p_ was obtained from the law, (MG/Area)^2^ = K_P_ × x × t.

Regarding the undeposited Ni superalloy exposed to the AO domain, Figure 9 indicates that it experienced a weaker whole mass gain (MG) and had a lower parabola curve rate constant (K_p_) value than the HVOF-deposited Ni superalloy. The mass gain at the fiftieth cycle for all the base alloys is tabulated in Table 3. A greater MG was seen under the MSE domain for both Ni superalloys (undeposited and deposited) compared to the AO domain. It is also revealed in Figure 9 and Table 3 that the HVOF-deposited base alloy subjected to MSE showed a lower MG and K_p_ value than the undeposited Ni superalloys. Figure 9 also illustrates that the MG quickly increased up to the eighth cycle in both Ni superalloys under both domains. After the eighth cycle, a continuous rise in MG was seen until the fiftieth cycle under both domains. The undeposited and HVOF-deposited substrates in both the AO and MSE environments conformed to the parabolic law for the corrosion rate (refer to Figure 10). K_p_ was obtained from the law, (MG/Area)^2^ = K_P_ × x × t.

### 3.5. Discussion of HT Corrosion on APS and HVOF Depositions

APS-deposited Ni superalloys revealed a high mass gain under the AO domain, which was ascribed to insufficient oxidation resistance as deposition decomposes into carbides when oxygen rejoins with Cr_3_C_2_ to produce Cr_2_O_3_ and releases CO/CO_2_. This will severely spall the deposited layers along edges, ascribed to the CO/CO_2_ pressure in deposition; moreover, thermal expansion differences exist between deposition and oxide formation. The generation of Cr_7_C_3_ and Cr_23_C_6_ carbide phases also observed when plasma sprayed Cr_3_C_2_NiCr coatings were subjected to HT (871 °C). In the APS-deposited Ni superalloys, only major elements such as Cr and Ni elements were found, which formed salt reactions with NiO and Cr_2_O_3_ oxides during the HC kinetics in the MSE.

During the first cycle of the HC test, oxidation of the undeposited Ni superalloy formed NiO at 400 °C; moreover, between 500 and 600 °C, Cr_2_O_3_ was formed under the MSE domain with the solid salt coating. Below 950 °C, the formation of non-shielding salt compound NaNiO_2_ and sulfate compound NiSO_4_, whose boiling point is 840 °C occurred [40]. At 900 °C, NaVO_3_ has a molten-stage temperature of 610 °C in a liquid state and begins to attack the thin protecting oxide films (NiO and Cr_2_O_3_) developed under salt accumulation [38]. They form the Na_2_CrO_4_ salt compound in the undeposited Ni superalloy under MSE, which evaporates in gas form at 900 °C, leading to chromium reduction and hence the reduced HR resistance, as noted by [41]. The HC resistance of the undeposited metal was better than that of the APS-deposited Ni superalloys, and this was ascribed to the formation of NiCr_2_O_4_, which is less porous [33].

The formation of NiO, Cr_2_O_3_ and NiCr_2_O_4_ oxides on the HVOF-deposited Ni superalloys was observed when exposed to the MSE domain. The high Cr presence in deposition delayed the hostile reactions of acid instability phase HC by developing well-built Cr_2_O_3_, which reduces the presence of V_2_O_5_ in MSE [33]. Improved oxidation resistance was seen in the undeposited rather than the HVOF-deposited Ni superalloys, as the oxide states (NiO, Ni_2_O_3_, and Cr_2_O_3_) and high-level uniformity were established to act as a dispersion barrier, which regulates the downward distribution of O_2_ into the Ni superalloy’s surface [25]. In the AO domain, the formed Cr_2_O_3_ and NiCr_2_O_4_ acted as diffusion barriers at the junction of the deposited surface under the AO domain to limit additional oxidation of the sublayers of the depositions. The growth of the Cr_7_C_3_ segment and unoccupied carbon in HVOF deposits was attributable to the breakdown of Cr_3_C_2_ [42]. The oxidation of Cr_3_C_2_ and Cr_7_C_3_ phases under the AO domain produced a mass gain in the HVOF-deposited Ni superalloy [39]. The low mass gain of the HVOF-deposited NI superalloy was due to the presence of the high quantity of Cr in the depositing powder’s composition. The high quantity of Cr_2_O_3_ decreased the ion matter of Na_2_SO_4_ to a very minimal value, at which sulfidation occurs.

## 4. Conclusions

The Cr_3_C_2_25NiC_r_ powder was coated successfully by the APS and HVOF techniques and showed a dense laminar structure with a bond/adhesion strength of more than 35 MPa and 75 MPa, respectively. The generation of protecting oxides such as NiO and Cr_2_O_3_ is responsible for the HC resistance of the undeposited and APS-deposited base metals in both corrosive environments at 950 °C. HVOF developed well-bonded coatings that could withstand fifty corrosion cycles at 950 °C without stripping off the base metal. The K_P_ value from the (mass gain)^2^ curve of the APS deposits is very close to the Ni superalloy under both AO and MSE conditions. APS-deposited samples successfully safeguarded the base metal alloy 80A against HC at 950 °C in MSE. The K_P_ value of the deposited superalloy is very close to that of the undeposited Ni superalloys in the MSE condition. When exposed to MSE, the collective weight gain in the HVOF-coated base metal was 7.30 mg/cm^2^ which is less than that of the undeposited alloy 80A, of 7.46 mg/cm^2^.

Based on the present investigation, the HVOF process shown to provide better coating characteristics in terms of bond strength and HT corrosion resistance in both environments compared to the APS process. HVOF deposits can be considered to protect base metals under MSE (Na_2_SO_4_ + 60% V_2_O_5_).

In recent years, the high-velocity air fuel (HVAF) process has emerged as an alternative to the HVOF technique for the deposition of carbide and metallic coatings, as it has high velocity and temperature, a particle size of 5 to 38 μm (15 to 45 μm in HVOF) and, in air, it reduces the oxidation of particles. Studies on the HVAF process with NiCrMoNb and Cr_3_C_2_-25NiCr coating powders are to be carried out under both AO and MSE conditions in the future. The HC resistance of Cr_3_C_2_-50NiCrMoNb, Cr_3_C_2_-40NiCr, N_i_CoCrAlH_f_Si and N_i_CoCrAlTaReY coating powders can be evaluated at 950 °C to determine whether they can be used as alternatives for Cr_3_C_2_-25NiCr powder under both AO and melted salt (Na_2_SO_4_ + 0.6V_2_O_5_) environments.

## Figures and Tables

**Figure 1 materials-14-05119-f001:**
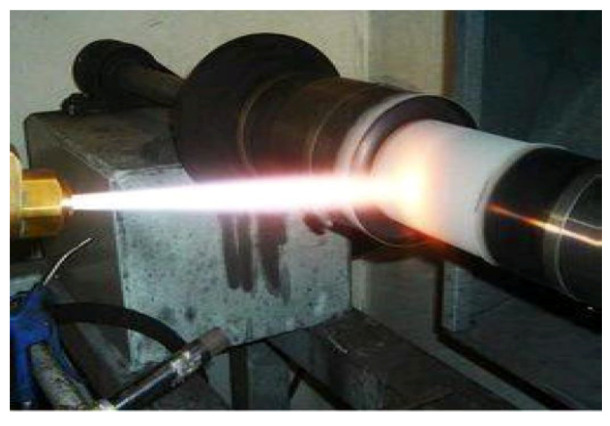
Photograph of plasma spray system.

**Figure 2 materials-14-05119-f002:**
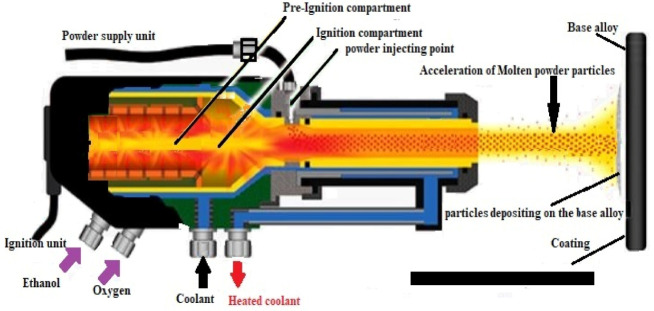
Schematic layout of HVOF spray system.

**Figure 3 materials-14-05119-f003:**
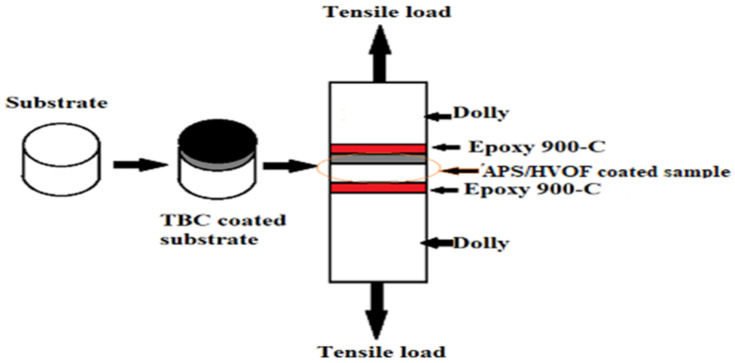
Schematic layout of bond strength test (ASTM C633) [34].

**Figure 4 materials-14-05119-f004:**
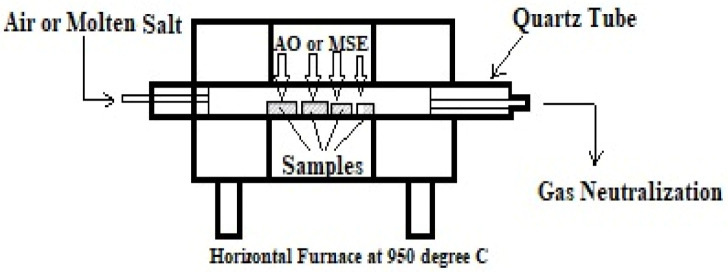
Schematic layout of HC test on samples in the tubular furnace subjected to the AO and MSE environments.

**Figure 5 materials-14-05119-f005:**
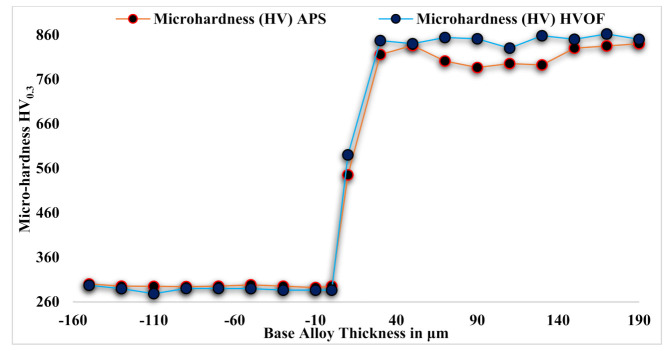
Microhardness of Cr_3_C_2_-25NiCr powder-coated (APS and HVOF) and uncoated base alloy.

**Figure 6 materials-14-05119-f006:**
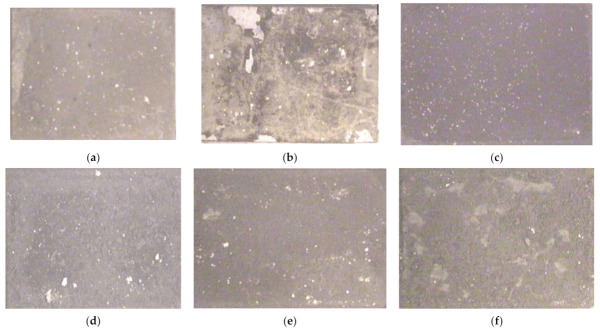
(**a**) Base metal under AO; (**b**) Base metal under MSE; (**c**) APS-coated sample under AO; (**d**) APS-coated sample under MSE; (**e**) HVOF-coated sample under AO; (**f**) HVOF-coated sample under MSE.

**Figure 7 materials-14-05119-f007:**
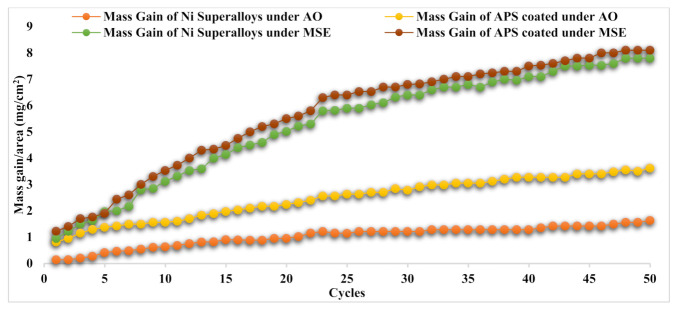
Mass gain vs. fifty corrosion cycles for the Ni superalloys (deposited and undeposited by APS) under both AO and MSE environments at 950 °C.

**Figure 8 materials-14-05119-f008:**
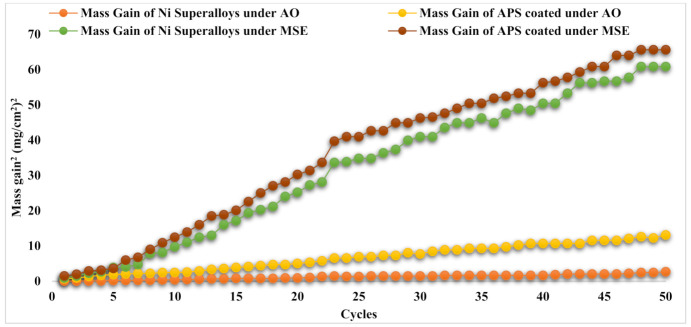
(Mass gain/area)^2^ vs. fifty corrosion cycles for Ni superalloys (undeposited and APS-deposited) under AO and MSE at 950 °C.

**Figure 9 materials-14-05119-f009:**
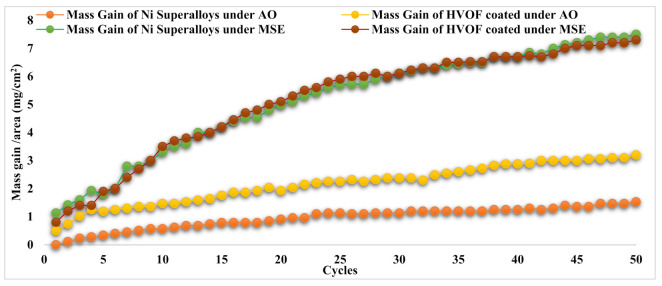
Mass gain vs. fifty corrosion cycles for the Ni superalloys (deposited and undeposited by HVOF) under both AO and MSE environments at 950 °C.

**Figure 10 materials-14-05119-f010:**
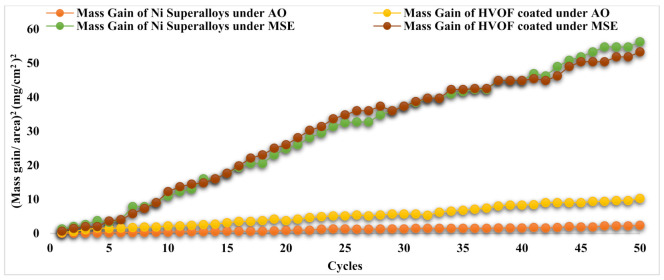
(Mass gain/area)^2^ vs. fifty corrosion cycles for Ni superalloys (undeposited and HVOF-deposited) under AO and MSE at 950 °C.

**Table 1 materials-14-05119-t001:** Chemical arrangement (in percent weight) of alloy 80A base metal and coating powders.

Base Metal and Coating Powder	Chemical Composition (Wt %)										
	Ni	Cr	C	Fe	Mo	Co	Nb	Al	Ti	Mn	Si
Alloy 80A base metal	Bal.	19.33	0.074	0.3	-	0.2	Nil	1.43	2.38	0.6	0.1
Cr_3_C_2_-25NiCr	20.60	Bal.	9.6	0.15	-	-	-	-	-	-	-

**Table 2 materials-14-05119-t002:** Control parameters of APS and HVOF spray coating technique for Cr_3_C_2_-25NiCr powder.

Process Parameter	APS	HVOF
Argon (Ar) flow rate (lpm)	55	-
Hydrogen (H_2_) flow rate (lpm)	12	-
Argon carrier flow rate (lpm)	3.2	8
O_2_ flow rate (lpm)	-	840
Jet fuel flow rate (lpm)	-	20
Ar, H_2_ and Ar carrier pressure (MPa)	0.5,0.5 & 0.5	-
O_2_, fuel and Ar carrier pressure (MPa)	-	1, 0.7 & 0.25
Powder inputting rate (gm/s)	0.66	0.58
Current (Amps)	600	-
Voltage (Volts)	80	-
Deposition angle	90°	90°
Stand-off distance (mm)	130	365

**Table 3 materials-14-05119-t003:** Total MG and K_P_ for undeposited and APS- and HVOF-deposited Ni superalloy under AO and MSE at 950 °C.

Specimen	Whole MG (mg/cm^2^)	K_P_ (MG/Area)^2^/Cycle, g^2^/cm^4^/s^1^
Undeposited Ni superalloy under AO	1.46	1.08 × 10^−6^
Undeposited Ni superalloy under MSE	7.46	2.78 × 10^−5^
APS-coated Ni superalloy under AO	3.35	5.03 × 10^−6^
APS-coated Ni superalloy under MSE	7.57	2.62 × 10^−5^
HVOF-coated Ni superalloy under AO	3.09	4.36 × 10^−6^
HVOF-coated Ni superalloy under MSE	7.30	2.5 × 10^−5^

## Data Availability

The data underlying this article will be shared on reasonable request from the corresponding author.

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
