# Peer review of "High-Temperature Corrosion of APS- and HVOF-Coated Nickel-Based Super Alloy under Air Oxidation and Melted Salt Domains"

_materials, 2021, doi:10.3390/ma14185119_

Round 1
Reviewer 1 Report
In this study, the authors investigated the high temperature corrosion behavior of APS and HVOF sprayed Cr3C2-NiCr coatings on Ni superalloy. However, the manuscript has the following problems:
(1) The novelty of the work is not clearly addressed in Introduction section. That is to say, why the Cr3C2-NiCr coating is worthy investigation, and it is strongly recommended to better introduce the reason for the interest on Cr3C2-NiCr coating compared to the existing available coatings fabricated by APS and HVOF.
(2) In the experimental section, the preparation parameters play an important role in the microstructures and properties of the coatings. Please, explain why you chose the parameters of APS and HVOF spraying.
(3) In the experimental section, the repeatability of the displayed results should be clarified. How many samples were used in the bond strength, cyclic high-temperature corrosion and micro-hardness tests should be explained. Otherwise, the readers cannot judge if the result will, or how frequent, emerge again.
(4) In the experimental section, Please explain the hardness measurements in more detail. How many indents were used per samples? Were any indentations rejected? If so, how many and why?
(5) The results from this study indicate proposed solution to important material damage problem. Moreover, the practical application of the protective coating method must be brought to the forefront in the conclusion part, in synchronization with the results obtained.
Author Response
Response to Reviewer 1 Comments
(1) The novelty of the work is not clearly addressed in Introduction section. That is to say, why the Cr3C2-NiCr coating is worthy investigation, and it is strongly recommended to better introduce the reason for the interest on Cr3C2-NiCr coating compared to the existing available coatings fabricated by APS and HVOF.
Reply: Added in the last section of introduction.
(2) In the experimental section, the preparation parameters play an important role in the microstructures and properties of the coatings. Please, explain why you chose the parameters of APS and HVOF spraying.
Reply: The APS and HVOF spraying parameters chosen based on the previous standards to provide standard form of composite coatings which forms defect free improved surface and minimizes other preparations required after coating to obtain the improved properties.
(3) In the experimental section, the repeatability of the displayed results should be clarified. How many samples were used in the bond strength, cyclic high-temperature corrosion and micro-hardness tests should be explained. Otherwise, the readers cannot judge if the result will, or how frequent, emerge again.
Reply: Added in the section 2.1.1, 2.2.2 and 2.1.3 and 2.1.4.
(4) In the experimental section, please explain the hardness measurements in more detail. How many indents were used per samples? Were any indentations rejected? If so, how many and why?
Reply: Explained in section 2.2.5 and section 3.1.
(5) The results from this study indicate proposed solution to important material damage problem. Moreover, the practical application of the protective coating method must be brought to the forefront in the conclusion part, in synchronization with the results obtained.
Reply: Carried out.
Reviewer 2 Report
This article faces a topic of interest as it is the stability of Cr3C2 based HVOF and APS coatings to hot corrosion in presence of molten salts. Nevertheless, the current state of the article is not correct and the work needs to be deeply reviewed in order to be accepted for publishing.
Introduction must be rewritten and shortened for clarifying the actual scope of the study.
In experimental section many information is missed. Many space is dedicated to explanation of HVOF and APS coatings preparation, however, low time is dedicated to explanation of corrosion tests. In Figure 3, why is included TBC coating in the scheme? Could you explain why do you use the term "dummy surface"?
You should show and explain how you are going to measure mass gain and the procedure for calculating Kp.
Results and discussion section do not sound totally convincible. I suggest you to join both sections and analyze both parts together. In some Figures of the results sections, legends are not complete.
Author Response
Reply to Reviewer2 comments
Introduction must be rewritten and shortened for clarifying the actual scope of the study.
Reply: carried out
In experimental section many information is missed. Many space is dedicated to explanation of HVOF and APS coatings preparation, however, low time is dedicated to explanation of corrosion tests. In Figure 3, why is included TBC coating in the scheme? Could you explain why do you use the term "dummy surface"?
Reply: corrosion test is explained in section 2.1.4. with Figure 4. Figure 3 is corrected. Dummy surface is also known as uncoated specimen with proper roughness to act as base or support in pull off test to determine the bond strength.
You should show and explain how you are going to measure mass gain and the procedure for calculating Kp.
Reply: Explained in section 2.13. about mass gain and procedure for Kp is high lightened.
Results and discussion section do not sound totally convincible. I suggest you join both sections and analyze both parts together. In some Figures of the results sections, legends are not complete.
Reply: Other reviewers do not want to be joined and figures legend sections are completed.
Reviewer 3 Report
The paper is focused on study new types of coatings resistant to corrosion. The topic of the work is fully in the field of interest for the readers of the journal and fully in the scope of the journal. The paper is well presented and can be considered for publication after a number of issues will be addressed. The points of concern are listed below:
- The Introduction has two points that need to be addressed: it is very lengthy (more than 35 % of the paper is Introduction) and also the focus of the Introduction is not well defined.
- Table 1 and Fig. 5, Fig.7-10 are missing the statistics of errors. What are the error bars in the measurements and how do they define them? In general, the analysis of errors of the experiments is not presented.
- The authors have to be more precise with the statements. See e.g. conclusions section “may give good results” – can you be more specific: what kind of good results? Is it referred to corrosion resistance or oxidation or something else? Such unclear statements can be found in different places in the text and the authors have to check the manuscript and fix the statements.
- The authors have to pay attention to the language as very often it is hard to read and understand the text due to the low quality of language.
Author Response
Reply to Reviewer 3 comments
- The Introduction has two points that need to be addressed: it is very lengthy (more than 35 % of the paper is Introduction) and also the focus of the Introduction is not well defined.
Reply: Modified with clear focus.
- Table 1 and Fig. 5, Fig.7-10 are missing the statistics of errors. What are the error bars in the measurements and how do they define them? In general, the analysis of errors of the experiments is not presented.
Reply: Error has been minimized by taking average of ten readings on each specimen was recorded as data point.
- The authors have to be more precise with the statements. See e.g. conclusions section “may give good results” – can you be more specific: what kind of good results? Is it referred to corrosion resistance or oxidation or something else? Such unclear statements can be found in different places in the text and the authors have to check the manuscript and fix the statements.
Reply: Attended.
- The authors have to pay attention to the language as very often it is hard to read and understand the text due to the low quality of language.
Reply: Improved through native English speaker.

Round 2
Reviewer 1 Report
I have accepted the authors' responses to all my comments. Some would require a long discussion, however, I take the style presented in the paper. In light of this revised manuscript is recommended for publication on the journal.
Author Response
Thanks for your valuable suggestions. We have modified the paper by reducing introduction part and editing english language.
Reviewer 2 Report
.
Author Response
Thanks foryour valuable suggestions. We have carried out english language editing and revised conclusion, and reduced introduction part.